# Statistical and Proactive Analysis of an Inter-Laboratory Comparison: The Radiocarbon Dating of the Shroud of Turin

**DOI:** 10.3390/e22090926

**Published:** 2020-08-24

**Authors:** Paolo Di Lazzaro, Anthony C. Atkinson, Paola Iacomussi, Marco Riani, Marco Ricci, Peter Wadhams

**Affiliations:** 1Italian National Agency for New Technologies, Energy and Sustainable Economic Development (ENEA), Dipartimento FSN, Centro Ricerche Frascati, via E. Fermi 45, 00044 Frascati, Italy; 2Department of Statistics, London School of Economics, London WC2A 2AE, UK; a.c.atkinson@lse.ac.uk; 3Istituto Nazionale di Ricerca Metrologica (INRIM), 00135 Torino, Italy; p.iacomussi@inrim.it; 4Dipartimento di Scienze Economiche e Aziendale and Interdepartmental Centre for Robust Statistics, Università di Parma, 43125 Parma, Italy; 5Independent Researcher, Via Fra Dolcino 19, 28100 Novara, Italy; tanoemarco@yahoo.it; 6Department of Applied Mathematics and Theoretical Physics, University of Cambridge, Wilberforce Road, Cambridge CB3 0WA, UK; pw11@cam.ac.uk

**Keywords:** statistical methods, measurement and error theory, inter-laboratory data comparison, robust statistics

## Abstract

We review the sampling and results of the radiocarbon dating of the archaeological cloth known as the Shroud of Turin, in the light of recent statistical analyses of both published and raw data. The statistical analyses highlight an inter-laboratory heterogeneity of the means and a monotone spatial variation of the ages of subsamples that suggest the presence of contaminants unevenly removed by the cleaning pretreatments. We consider the significance and overall impact of the statistical analyses on assessing the reliability of the dating results and the design of correct sampling. These analyses suggest that the 1988 radiocarbon dating does not match the current accuracy requirements. Should this be the case, it would be interesting to know the accurate age of the Shroud of Turin. Taking into account the whole body of scientific data, we discuss whether it makes sense to date the Shroud again.

## 1. Introduction

The Shroud of Turin is a linen cloth measuring 4.4 m by 1.1 m bearing the faint imprint of head-to-head, frontal and dorsal body images of a man with apparent wounds, laid out in death as if the images had been formed while the cloth was longitudinally folded over a human body. The positions of bloodstains are consistent with the specific details of Jesus’ crucifixion: the thorns on the head, the scourging, the nail wounds on feet and wrists, and the wound in the side. There are many triangular holes with charred edges made by a documented fire of 1532. Several water stains are also visible.

After conducting the most extensive, multidisciplinary scientific examination of the Shroud in 1978, a team of scientists under the auspices of the Shroud of Turin Research Project (STuRP, see https://www.shroud.com/78team.htm) published the main results in 30 papers, listed in https://www.shroud.com/78papers.htm. The most important result of the STuRP analyses is that the 0.2-micrometer-thick sepia color of the body image was produced by a process related to oxidation, dehydration, and conjugation of the polysaccharide structure of the microfibrils of the linen itself [1,2,3]. The image is unlikely to have been painted; it is difficult to create an image that matches the complex chemical and physical characteristics of the image at the microscopic level and its peculiar superficiality [4]. According to the STuRP final report “*the Shroud image is that of a real human form of a scourged, crucified man. It is not the product of an artist*”. Recent experiments [5,6] and [7] confirmed the difficulty of replicating the superficial coloration processes of the linen cellulose at the microscopic level.

In 1988, a peripheral piece of the cloth was radiocarbon dated to the 14th century by Accelerator Mass Spectrometry (AMS) [8]. The technique of radiocarbon dating relies on the fact that living plants and animals contain carbon isotopes, namely, the stable ^12^C and ^13^C and the radioactive ^14^C, in a ratio close to that of the Earth’s atmosphere. From the moment of death, if no external contamination occurs, the organism becomes a closed system in which the stable isotopes ^12^C and ^13^C maintain their concentration, while the amount of ^14^C decreases at a known rate. By measuring how much ^14^C an artifact contains, analysts determine the “radiocarbon age” of the sample. The true “calendar age” is obtained by combining the radiocarbon age with a calibration curve which takes into account the variable concentration of ^14^C due to fluctuations of, e.g., solar wind and the Earth’s magnetic field, as well as to nuclear weapon testing and ^14^CO_2_ cycling between atmospheric, oceanic, and terrestrial carbon reservoirs. An overview of the metrological history of ^14^C dating can be found e.g., in [9].

The results of [8], which claimed that the cloth is mediaeval, support the suggestion that the image was produced by an artist (see for example [10]). A referee of this paper pointed out that “*the work appeared around 1353–1354 in a new collegiate church in Lirey (in the Savoy Alps) and its provenance was the subject of detailed discussions at the time, as noted by two bishops of the region*”.

In this paper, we review the results of the radiocarbon dating of the Shroud of Turin, in the light of robust statistical analyses of published data and a recent examination of raw data. Our purpose is to clarify the statistical analysis, leaving aside historical speculations on the origin of the Shroud.

These analyses reveal an inter-laboratory heterogeneity of the means and an anomalous systematic spatial gradient of the ages which make it incorrect to combine the results of the three laboratories into a single overall estimate of age. Taking into account the whole body of the scientific data, we discuss whether it does make sense to date the Shroud again.

The paper is organized as follows. The overall scenario, sampling, and results of the 1988 Shroud dating are traced in Section 2. Section 3, Section 4 and Section 5 are devoted to robust statistical analyses of published and raw radiocarbon data, and the design of statistically correct sampling procedures. Section 6 explains why dating textiles need specific attention vs. other materials, and Section 7 faces the basic question: Is it possible to determine the reliability of the radiocarbon age of the Shroud taking into account the influence of contaminants? Concluding remarks are presented in Section 8.

## 2. Radiocarbon Dating the Shroud: Context, Sampling, and Results

In 1984, STuRP proposed an extensive program to re-examine the Shroud, to gain a deeper insight into the image formation, the age of the cloth, and to find the best conditions for long term conservation. Out of 26 tests proposed, the Pontifical Academy of Science suggested that the Holy See accept only one, the radiocarbon test. In 1986 the “Turin Workshop Protocol” stated that seven institutions were chosen to perform two ^14^C dating techniques, namely, the proportional counter and the AMS.

The radiocarbon measurement is destructive, as the sample to be dated is burned. This is the main reason this protocol was modified to reduce the amount of cloth to be sacrificed. Only the method of AMS would be used and just three laboratories were selected: the Laboratory for Archaeology and History of Art, University of Oxford, the Department of Physics, University of Arizona, and the Institute für Mittelenergiephysik, Zurich. The British Museum was the only advisor for the proper execution of sampling and dating. In the last meeting, held at the British Museum on 22 January 1988, each laboratory asked for 40 mg of cloth as the minimum weight necessary for reliable measurements [11]. The main changes in the protocol were summarized by [12].

The changes in the protocol caused controversy. The director of the Rochester Laboratory, one of the four not selected, wrote a seven-point letter to Nature criticizing the new protocol. Point one reads: “*The involvement of seven laboratories has been reduced to three. This eliminates the possibility of detecting a mistake made in the measurements by one or more of the three laboratories. As Tite knows, such mistakes are not unusual*” [13].

This concern was reasonable in the 1980s because, in some cases, a combination of at least two techniques was indispensable for the highest level of confidence. Indeed, the results of international radiocarbon inter-comparisons [14,15] showed that the accuracy and reliability of AMS in the late 1980s were considerably less than today [16].

Despite the above concerns, on the 21 April 1988, the Shroud was separated from its backing cloth along the bottom left-hand edge of the frontal image and a strip (~8 cm × 2 cm) was cut from the edge of the cloth adjacent to a sample removed by G. Raes of the Ghent Institute of Textile Technology in 1973, see Figure 1 and Figure 2.

The single site was chosen to minimize the visual impact of the cut fabric. However, the use of a single sample, assuming it was representative of the whole cloth, defied normal sampling procedures.

The strip was divided into four parts: one, “Riserva”, was retained by the Archbishop of Turin and three were to be dated; see Figure 2. Because the sub-sample given to Arizona was about 13 mg smaller than the others (39.6 mg Arizona vs. 52 mg Oxford and 52.8 mg Zurich) they were given a second 14.1 mg sample cut from the Riserva, as shown in Figure 2.

The results were announced at two press conferences held on 13 October 1988 in Turin and London, and the technical details were published four months later in [8]. The three averaged conventional radiocarbon years *before present* (BP)—where “present” refers to 1 January 1950 AD—were: Arizona = 646 ± 31; Zurich = 676 ± 24; Oxford = 750 ± 30. A confidence interval was calculated for the radiocarbon age, using the *t*_5_ distribution, that is, the t-statistic with 5 degrees of freedom. This was then made slightly broader to allow for the errors in the calibration curve before using the curve to give the calendar age. The details are on p. 614 of [8]. In this respect, we note that a consolidated and globally agreed approach/document on uncertainty evaluation was not available in 1988. In [8] uncertainty and errors are evaluated using a method no longer implemented by metrologists. In fact, during the 1980s, the approach to measurement uncertainty was based on random and systemic error statements. The Bureau international des Poids et Mesures at the end of the seventies launched a questionnaire on the evaluation of uncertainty. National Metrology Institutes highlighted the need for guidelines to tackle the uncertainty evaluation method discrepancies, merging the approaches and starting work for the first edition of the Guide for expression of Uncertainty in Measurements (GUM) in 1993, see Section 5.

After the calibration, the BP years were translated into the calendar age, resulting in a double range: 1262–1312 AD and 1353–1384 AD. Joining the two probability curves, one gets 1262–1384 AD, which, rounded to the decade, gives the sample age between 1260 and 1390 AD, with a confidence level of 95%.

Some scholars have tried to evaluate the reliability of the Shroud dating results, and speculation ensued on possible skewing factors [18,19,20,21,22].

Experimental evidence of a bad choice of the sampling site was pointed out in [20] by chemical analyses of threads of the radiocarbon sample, and in [23,24], by Fourier Transform Infrared Spectroscopy (FTIR) of fibers from three threads of the pieces trimmed off the radiocarbon sample (see the top drawing of Figure 2). The FTIR identifies the chemical structure of a molecule by its infrared absorption spectrum which is a fingerprint able to identify molecules. The analyzed fibers had a different FTIR spectrum vs. non-image fibers removed by STuRP from the bulk of the Shroud. According to [23], the FTIR data for the radiocarbon sample shows physical characteristics of both the water stain and scorch regions of the cloth. As a consequence, FTIR spectra show the area selected for the radiocarbon sampling is atypical and is not representative of the rest of the Shroud.

Additional remarks on the reliability of the dating results were related to statistical studies of the official data by [25,26,27], which raised several concerns, including statistical tests which do not support the 95% confidence level of the range 1260–1390 AD, and statistically significant differences between the data reported by the laboratories in both the mean values and the error terms. A common conclusion of the statistical studies is that the measured samples are not homogeneous, and the means are not compatible with each other. As a consequence, the radiocarbon dates are not reliable, and the measurements should be repeated. Unfortunately, some concerns could not be confirmed or rejected because the three laboratories refused to release the raw data, including the position of each subsample.

## 3. Robust Statistical Analysis of the Official Data

A major improvement in understanding the reliability of the Shroud’s radiocarbon dating occurred in 2010 when robust methods of statistical analysis were applied to the 387,072 possible configurations of the unknown position of the 12 official subsamples in [8]. This analysis determined the most probable position of each subsample, finally leading to the following results [28,29]:(i)The smaller one of the two Shroud samples given to the Arizona laboratory was not dated;(ii)The data published in [8] are heterogeneous and there is a linear spatial gradient of the 12 subsample ages. That is, the age of a piece at the top edge is systematically less than that of the adjacent piece. As a consequence, the subsample dating cannot be considered as repeated measurements of a single unknown quantity. Thus, the basic assumption of radiocarbon dating was not fulfilled.

Point (i) was surprising, because [8] did not report that only three samples out of four were dated in 1988. Everybody took it for granted that the dating was the result of the ^14^C counts of four samples.

On December 2010, the Arizona AMS Laboratory showed in [30] recent photographic images of a Shroud sample never dated and still in possession of the Arizona laboratory. Later, it was clarified that the piece of linen shown by [30] is a remnant of the dated sample and that the smallest Arizona sample was never dated, thus confirming the finding of the robust statistical analysis.

### 3.1. Results

Robust statistics are not unduly affected by outliers. As a consequence, as well as providing desirable parameter estimates, they are helpful in identifying outlying observations and departures from model assumptions [31]. Here we describe the use of outlier detection from robust fitting in determining the spatial location of the samples analyzed by each laboratory, thus expanding the results in [29].

Table 1 gives the estimated radiocarbon BP years of the official 12 sub-samples of the Shroud as reported in [8]. The table also shows the 68% confidence limits (1.1 standard errors based on the *t*_5_ distribution), which is considered the estimated uncertainty of the individual measurements, as stated by [8].

As a first step in clarifying the nature of the heterogeneity in the data, we apply a traditional ‘Analysis of Variance’ (ANOVA, a collection of statistical models and their associated estimation procedures used to analyze the differences among group means in a sample) to verify whether the results of the three laboratories can be combined. If so, the 12 BP years can be considered as repeated measurements of a single unknown quantity. The 1989 analysis solely performed a test (see Table 2 of [8]) on the homogeneity of the standard errors of the three laboratories. On the other hand, [28,29] performed ANOVA on the single published measurements to check the hypothesis of equality of means among the three laboratories. Since the conclusions of the test might be affected by the different standard errors across the different laboratories, [28,29] checked the stability of the results to allow for the potential underestimation of uncertainty of a particular laboratory as well as possible differences among the protocols which have been used. They apply:(1)Unweighted ANOVA: we ignore the estimated uncertainty of the individual measurements;(2)Weighted ANOVA: we use the scaled standard errors in Table 1 as weights;(3)Modified ANOVA: If the weights are correct, the observed standard errors should agree with those given by the weights. They do not for Arizona, with the weights giving standard errors that are appreciably smaller than the observed values. We use the observed standard errors detailed in [8] to scale up the individual weights for Arizona.

Since the ANOVA test can only be correctly applied if the variability of the three laboratories is the same, we compute the Box test of homogeneity of variances [32]. Table 2 shows the results of the traditional ANOVA applied to the 12 calculated ages in Table 1.

The first line of Table 2 gives the significance levels for the Box test of homogeneity of variance, showing no evidence of non-homogeneous variance, whichever weights are used. The significance levels of the tests for the means are given in the second line of the table. All three tests are significant at the 5% level, with that for the original weights having a significance level of 0.0043, one order of magnitude different from that of the other analyses.

In summary, the ANOVA analysis of the dating results presented in [8] shows:No evidence of differences in variances among the three laboratories;Evidence of difference in means among the three laboratories.

Each laboratory also dated three control samples, namely, a linen textile from a Nubian tomb (XI-XII century AD), an Egyptian mummy from Thebes (110 BC-75 AD), and threads from a cope from Var, France (1290–1310 AD). None of the control samples was controversial. If we repeat the ANOVA analysis described above on the three control samples, we find:No evidence of differences in variances between the three laboratories.No evidence of a difference in means between the three laboratories.

To sum up, since the 12 official subsamples were taken from the same piece of cloth, the radiocarbon ages should have been equal within errors, but they were not. The dating of the three control samples confirmed that the AMS worked correctly for the control samples. Consequently, a hypothetical “laboratory effect” is to be excluded for the Shroud dating because the differences in means of the control samples are not significant among laboratories.

We now investigate the reasons for the lack of homogeneity of the Shroud data and discover the source of the heterogeneity in means.

It is stated in [8] that the Arizona laboratory dated four subsamples; the Oxford laboratory divided its piece into three parts, and Zurich into five parts. In [8] it is not explained how the laboratories subdivided the single samples received. In the absence of this information, we consider all the plausible ways in which the three laboratories may have divided the samples, ending up with 96 (Arizona) × 24 (Oxford) × 168 (Zurich) = 387,072 cases to analyze.

To detect any trend in the age of material (which could explain the lack of homogeneity) we can fit 387,072 regression models, where the response is the estimated age (as given in Table 1) and the coordinates of the two explanatory variables are given respectively by the longitudinal (horizontal in the upper panel of Figure 2) and transverse (vertical in the upper panel of Figure 2) distance.

We use t-statistics to test the significance of the estimated rate of change of age in the longitudinal and transverse directions. The robust statistical analysis reveals that:(1)The distribution of the t-statistic for the vertical coordinate is not centered around zero (as expected if the vertical coordinate did not play any role), but there is not enough statistical evidence to claim that it is significant. Given that the whole sample is rectangular with the long side horizontal, see the upper panel of Figure 2, we do not have enough information to detect significant variability along the vertical coordinate, which; however, cannot be ruled out.(2)The t-statistic for the horizontal component is always negative. It is not significant when Arizona is assumed to have dated both samples, but becomes significant if only the larger sample were dated, providing evidence of an anomalous and unexpected relationship, with a negative slope of about 50 years/cm, between age and horizontal position.(3)The configurations that assume Arizona dated both samples lead to regression models with outliers, which are absent under the assumption that Arizona dated just the larger sample. The latter allocations are those giving a significant slope [29].

The broad conclusions are that Arizona analyzed only one of the two samples and that the systematic spatial gradient of the ages explains the difference in means that was detected by the ANOVA of [29], summarized in Table 2, and recently confirmed by the analysis of the raw data as detailed in Section 4. The analysis of [8] did not consider differences in means.

### 3.2. Implications of the Evidence of a Spatial Trend

One of the referees asked for clarification of the contribution and implication of the new ANOVA combined with the presence of a spatial trend because the original reference [8] had noted that the chi-squared value for the Shroud sample indicated that the errors might have been underestimated.

In the former ANOVA we show that the results about the lack of homogeneity of the means do not depend on whether or not one takes into account the underestimation of the uncertainly of a single laboratory.

In addition, if the means are homogeneous and there is underestimation of the errors of one lab, a summary of the results through a weighted average is methodologically correct. On the other hand, the combination of the results into a unique age with a unique confidence interval is not meaningful in the presence of significant linear trend, that is in the presence of spatial non-stationarity. Once again, the presence of a spatial trend does not show that the Shroud is mediaeval or is not mediaeval, but simply clarifies what can be claimed from this analysis and what cannot be claimed. In other words, in order to avoid misunderstandings, we do not claim that the radiocarbon results are incorrect, and it is not our purpose to try to shift the time range. Here we simply point out that in general the results coming from different sources can be legitimately combined together even if there is different accuracy among the different protocols, but not if the results they produce depend on different systematic characteristics (such as the position of the raw elements where they were taken). More about this topic can be found in the response to referees section attached to this paper.

## 4. Statistical Analysis of the Raw Data

The next important step in understanding the reliability of the Shroud’s radiocarbon dating occurred in 2017 when a legal request based on the Freedom of Information Act in the United Kingdom compelled the British Museum to release the raw data of the radiocarbon dating of the Shroud. The raw data were analyzed and, according to the final report [33], the results do not validate the 95% confidence level claimed in [8], as the data contain “*serious incongruities*”. Interestingly, the overall analysis of the raw data confirms the main results in [29].

### 4.1. Results

The file which includes the raw data kept by the British Museum consists of 711 unpublished pages, showing that, contrary to the report in [8], Zurich performed ten measurements, Oxford performed five measurements, and while the detailed measurements were not sent to the British Museum, a mean was provided for two measurements, resulting in five measurements and three radiocarbon dates (2 + 2 + 1), as reported in Table 1. Arizona divided the sample ‘Arizona 1’ into three samples of 12.39 mg, 14.72 mg, and 11.83 mg. The first 12.39 mg sample was retained for possible future checks. Each of the two remaining samples was subdivided into two, making a total of four sub-samples for dating. Each sub-sample was processed separately, and in each case the resulting graphite was radio-dated twice, resulting in eight independent determinations, not four as reported in [8], see Table 3.

Using the raw data, parametric (ANOVA) and non-parametric tests (OxCal diagnostic tool, https://c14.arch.ox.ac.uk/oxcal.html) were used to determine both intra- and inter-laboratory differences between means in the raw data compared with the 12 official data sets, see Table 1.

The results show that inside each laboratory, there is no statistical evidence of significant disagreement between the original measurements and those published in [8]. On the other hand, when using the raw data to validate the homogeneity of the results produced by the three laboratories, both indexes of agreement of the individual measurements are much below the required threshold, and the small *p*-values of the test by [34] show that there are noticeable differences among the raw data of different provenances. These differences remain significant after applying conservative corrections for multiple tests [35]. While the measurements inside each laboratory seem to be homogeneous, the analysis of the raw radiocarbon dates confirms that different laboratories produced different assessments that are statistically significant.

Thus, the analysis of the raw data confirms the heterogeneity of the means produced by the three laboratories as described in Section 3. In particular, the consistency between the Arizona raw radiocarbon dates and the published results from Zurich and Oxford is questioned. As a consequence, the measurements of the age of the Shroud cannot be considered as repeated measurements of a single unknown quantity.

These findings broadly agree with those in [25,26,27], mentioned in Section 2, and with those summarized in Section 3, which refer to the differences of means based on the 12 official measurements.

In summary, the analysis of the raw data shows that the hypothesis of a constant amount of ^14^C among the dated subsamples is rejected. The evidence is:(a)The heterogeneity of the raw data;(b)The consistent ages of the control samples.

According to [33], homogeneity is lacking in the data and the analyses show the necessity of further radiocarbon dating to compute a new reliable interval. This new test requires, in interdisciplinary research, a robust protocol.

In the next section, we suggest such a protocol for a sampling design able to provide accurate dating results which are representative of the whole cloth.

## 5. The Correct Sampling Design

A major difficulty in coming to a reliable dating of the Shroud is the unrepresentative way in which the sample was taken, see Figure 1. Experimental results described in Section 2 show that the linen of the chosen corner was somehow different or had different contamination vs. the main cloth. In this section, we briefly consider some correct ways of sampling. First, we suppose that samples could be taken from several sites of the cloth. These ideas are useful when we restrict ourselves to designs where samples are only taken from the edges.

If the radiocarbon age of the Shroud were expected to vary smoothly with horizontal and vertical distance, it would be appropriate to use an experimental design such as those in [36,37] to estimate this smooth response surface.

However, if the Shroud has been contaminated, the purpose of the design would be to take as many measurements as possible from uncontaminated areas. A space-filling design is then appropriate, such as those used in computer experiments [38]. In two dimensions, the ‘Latin-hypercube’ designs would be generated by conceptually dividing the Shroud into *n* rows and *n* columns, creating *n^2^* potential experimental units. A set of *n* units is then chosen for experimentation. The spatial cover is achieved by choosing the units such that there is one in each row and column. The units can be chosen at random, and any seemingly unsatisfactory pattern, such as one that is spatially too regular, rejected. Alternatively, sampling can be only from a set of units which have some desirable spatial property. [39] prefers the latter. Once the units for experimentation have been chosen, the samples need to be assigned to the three laboratories in a suitably randomized way to, e.g., avoid all samples assigned to a given laboratory coming from one end or side of the material, as actually happened with the Arizona laboratory. The design of spatial experiments is given in [40].

Until less invasive methods of dating are employed, samples will presumably be confined to the edges of the Shroud cloth to minimize the visual impact of the cut. However, the preceding discussion does provide guidance on a suitable design. If *n* samples are to be taken, the perimeter of the material should be divided into *n* intervals of as equal size as possible. Locations are then selected at random within each interval, preferably subject to a restriction on the minimum distance between samples. The intervals might also be chosen to exclude corners or edges of the material when it is suspected that these areas are more contaminated.

Suppose *n* = 12. Then 12 independent samples could be taken from the edges and ends. However, a good design should allow some internal estimates of inter-laboratory and random effects. A possibility is to sample 6 points and to divide each sample into two subsamples. Let us consider the case of the three laboratories in the 1988 dating; there are 3 possible pairs of allocations of the subsamples: AO, AZ, and OZ so each can occur twice, the allocation of treatments (pair of sites) being made at random, again guarding against any obvious spatial pattern. The resulting design is a balanced incomplete block design with two treatments per block. For a discussion of the desirable properties of such designs, see, for example, [41].

The difficulties in interpretation of the results of the age of the Shroud show forcibly the difficulties that can be caused by inadequate experimental design. Laboratory scientists are trained, perhaps subconsciously, to trust that they control all sources of variation. Statisticians, on the other hand, are aware of the possibility of unexpected sources of heterogeneity (here the “lurking variable” of location) which have to be guarded against by suitable randomization [42]. An example is the too-small values of the standard errors for the Arizona data in [8].

Furthermore, even if all possible sources of heterogeneity are avoided, a measurement uncertainty study must be performed, and careful records need to be taken of any possible other variables. The exact location of the subsamples that were inferred in Section 3 is an example.

In a future dating measurement, the design of a statistically correct sampling procedure is mandatory. Here we have detailed a statistical procedure that may help to avoid possible sources of heterogeneity.

## 6. Specific Problems of Textile Dating

The porous nature of cellulosic textiles makes it difficult to find virgin parts of the yarns which have never been in contact with polluting materials. The surface per unit weight of fabrics is much larger than most objects due to the small diameter of the fibers, of the order of 20 micrometers. Yarns and fibers of cellulosic fabrics are highly permeable to any liquid, gaseous and solid contaminants.

According to [8], the three laboratories used different preliminary cleaning techniques. Zurich treated all subsamples by the standard Acid-Alkaline-Acid procedure (referred to as AAA). Arizona cleaned two sub-samples by AAA, and two other subsamples by rinsing them with HCl and commercial detergents and then submitting them to Soxhlet extraction in ethanol. Oxford cleaned subsamples in petroleum ether before using the AAA procedure and finally bleached two of them in sodium hypochlorite. The different cleaning procedures may have contributed to the inter-laboratory heterogeneity of the means and the anomalous systematic spatial gradient of the ages pointed out in Section 3 and Section 4. When foreign substances infiltrate the textile forming chemically stable bonds with the cellulose, not all cleaning procedures are able to remove them by selectively breaking their bonds. There is a possibility of pollution which might escape pretreatment. Then, it is difficult to distinguish the contribution of foreign ^14^C from that originally present, and the overall ^14^C count adds the original radiocarbon to the ‘new’ one, thus skewing the dating result. As a consequence, the accuracy in the radiocarbon dating of a textile depends on its age, handling, and exposure to contaminants during its history. This problem was particularly serious in the 1980s, when contamination was a main contributory cause of the unsatisfactory results (three outliers out of 18 radiocarbon ages of three textile samples) obtained in inter-comparison work between four AMS and two gas counter laboratories [14]. The authors indeed ascribed the presence of outliers to inadequate cleaning procedures.

The need to pay attention to sampling and contaminants was confirmed in 1990, following an international radiocarbon inter-comparison involving over 38 laboratories worldwide. The challenge was to date a set of samples of known age by using gas proportional counting, liquid scintillation counting, or AMS. Of the 38, only 7 produced results that the organizers considered to be satisfactory [15]. The trial pointed out that the major sources of variability for the five AMS laboratories involved were the sampling and pretreatment processes, that is, the same problems revealed by the statistical analyses summarized in Section 3 and Section 4.

Today, the above concern is echoed in the webpage of Beta Analytic, a leading provider of radiocarbon dating services. According to Beta Analytic, standard acid/alkali/acid pretreatments are effective for the removal of carbonate and soluble humic acids. However, with textiles that have been handled greatly and exposed to human influences, special treatments such as solvent and cellulose extractions are recommended. After a standard acid/alkali/acid pretreatment, the sample is bathed in sodium chlorite under controlled conditions (pH 3 and temperature of 70 °C). This procedure eliminates all components except wood cellulose. At least 100 mg are needed for these additional pretreatments.

These considerations show that, nowadays, the radiocarbon community is aware of contamination issues, and that there are strategies to remove additives and preservatives. In this paper, we refer to the controls available in 1988, considering the severe pollution of the Shroud, detailed in the following.

### 6.1. Clean Samples of Polluted Cloth

The images of a sample retained by Arizona and analyzed by [30] show a relatively clean sample, with few cotton fibers and small amounts of foreign materials. According to the head of the Zurich laboratory [43], it was proved experimentally that none of Zurich samples was contaminated. Optical microscope and SEM images of the Oxford subsamples confirm the absence of heavy pollution. Thus, the three laboratories agreed they received reasonably clean samples.

In contrast, several analyses show that the Shroud fibers are dirty and heavily polluted by dust, burned shards, mucilage, mildew, spores, mites, and fungi, as commonly found in very ancient linen, like that of Egyptian mummies. As an example, SEM screening of the Shroud fibers performed by [44] showed “*abundant deposits of pollutant material extraneous but intimately connected with the individual fibers of the cloth. This type of dirt is not found in recent linen threads. The comparison with threads coming from an Egyptian mummy provided by the Egyptian museum of Turin gave an ultrastructural framework superimposable with that of the Shroud thread*.”

More recent SEM analyses of Shroud fibers in [45] confirmed the abundant deposits of pollutants reported in [44].

Interestingly, recent microscopy and SEM analyses of linen threads from a copy of the Shroud dating back to 1653 [46] reveal cleanliness similar to that of the fibers of the sample dated by Oxford shown in https://archdams.arch.ox.ac.uk/?c=1203&k=1bcdc90a8b. This similarity suggests a possible reason for the lack of heavy visible pollution of the radiocarbon sample: possibly, the sampling site included a patch added in later centuries, a repair having been made to the corner. This hypothesis is supported by the following results:(a)The analyses of threads from the radiocarbon test area by [20] and by the FTIR spectroscopic data for the radiocarbon sample by [23,24] show that the area of the radiocarbon sample is atypical and is not representative of the rest of the Shroud;(b)The presence of spurious cotton detected by microscopy [47] and by analytical spectroscopy in [48] inside threads from the Raes sample (which is adjacent to the radiocarbon sample, see Figure 2) might reveal an attempt to repair and reweave one of the portions historically removed from the Shroud by the Savoy family over the centuries for various reasons, as reported by [49];(c)According to [30], the thickness of the remnant of the Arizona sample is 0.25 mm, which is considerably less than the average 0.39 mm thickness of the Riserva (minimum 0.34 mm, maximum 0.43 mm) measured by the textile expert [50], thus supporting the hypothesis of the presence of a foreign fabric in the Arizona sample.

On the other hand, the textile expert [51] did not find discontinuities or mending visible to the naked eye in the area adjacent to the radiocarbon sampling.

In another respect, robust statistical analyses of the official data in Section 3 show an unexpected and anomalous relationship, with a negative slope of about 50 years/cm, between radiocarbon age and position. This is a systematic spatial effect possibly due to the uneven efficacy of cleaning pretreatments. In this frame, [52] proposed the hypothesis that the different cleaning pretreatments carried out by the laboratories (see Section 6) may have removed contaminants unevenly, causing a spatial gradient of residual contamination that could explain the relationship between the radiocarbon age and the position of dated subsamples.

## 7. Discussion and Proposal

Virtually all researchers agree that it would be interesting and methodologically informative to repeat the radiocarbon dating of the Shroud. Sufficiently small samples can now be measured so that the appearance of the relic would not be much altered by sampling different sites. However, any attempt to date a fabric without knowing the conditions in which it was stored and handled, carries a risk of obtaining an inaccurate age determination, as discussed in Section 6.

Today, we know that:(i)The anomalous relationship between subsample age and its position revealed by the robust statistical analysis (see Section 3.1) might suggest the presence of contaminants that escaped cleaning pretreatments. These contaminants could have biased the radiocarbon age;(ii)The use of additives or preservatives during the history of the Shroud is highly probable albeit poorly documented. We know that thymol was left in the wooden reliquary during the whole day of the 21 April 1988 when the Shroud was taken down for the sampling operation [53]. As a consequence, the Shroud was exposed to thymol vapors released by the wood from 1988 to 1998, when the Shroud was moved into an especially designed exposition reliquary.

Exposure to thymol (as well as to other preservatives) could increase the probability of obtaining inaccurate ages in a new radiocarbon dating. The reaction between the phenolic group of thymol and the cellulose hydroxy groups can take place without heat—or another form of energy—supplied from the outside because in the reaction there is little or no variation of enthalpy. In fact, the standard enthalpy of reaction for similar model compounds in the condensed phase is ΔH° ≈ 0 kJ/mol. Although unlikely in the absence of a strong acid as catalyst, such a reaction cannot be ruled out. Therefore, today a new ^14^C measurement of the Shroud could still result in an inaccurate age determination due to thymol exposure. To get valuable information in advance of any possible new dating, we propose a strategy to determine to what extent the influence of contaminants may skew the radiocarbon age of the Shroud, as detailed below.

### 7.1. Charred Material

On 4 December 1532, the Shroud sustained a fire, which scorched the folded fabric, creating several holes with charred edges. In the spring of 1534, the Poor Clare Nuns patched the holes and attached a full-size support cloth to the back of the Shroud. This became known as the Holland cloth. The possibility of dating the charred threads of the Shroud was discussed during the Turin Workshop (see Section 2) and it was suggested that the fire might have altered the ^14^C/^12^C ratio, possibly via isotopic exchange between the cloth and some volatile combustion products, as discussed by [19] and [54]. Since it is likely that any effect would have been proportional to the temperature of the fabric during the fire, a comparison between the radiocarbon ages of charred material and of unaffected Shroud threads should provide valuable information on this issue. Moreover, the burned threads come from several areas of the Shroud and are less porous than intact threads, thus less prone to being contaminated. Then, dating charred threads could give information on the spatial distribution of contaminants attached to the cloth before the 1532 fire.

### 7.2. Holland Cloth

When dating the Holland cloth, we expect a date compatible with the 16th century. Any different result would be a hint of non-removed contaminants which polluted the cloth from 1534, when the Holland cloth was sewn along the perimeter on the Shroud, until 2002, when it was removed during the restoration works. In this frame, the Holland cloth can be used as a benchmark, to assess the best pre-treatment procedure to obtain an accurate result.

### 7.3. Shroud Fibers and Raes Sample

Dating non-image fibers in sites far from both water stains and burnt areas can provide—by comparison with the radiocarbon dates of charred materials and Holland cloth—information on the effect of all Shroud contamination accumulated up to now, including preservatives. Dating the Raes sample—it was adjacent to the sample dated in 1988, see Figure 2—may provide information on the spatial gradient of contamination suggested by the robust statistical analyses in Section 3.

## 8. Conclusions

Although the Shroud of Turin is one of the most studied archaeological objects in history, the sum result of its study offers more questions than answers. On the one hand, the body image defies simplified explanations of the mechanism by which it was produced, see Section 1. On the other hand, the AMS technology in the 1980s provided a medieval dating of the linen cloth. However, when taking into account:(a)The single-site Shroud sampling procedure (see Figure 1), which does not protect against the possibility that the sample is not representative of the whole, as discussed in Section 2 and Section 5, as well as chemical and FTIR spectroscopic data for the radiocarbon sample, which showed the area of the radiocarbon test was atypical and not representative of the rest of the Shroud [20,23,24];(b)The difficult cleaning of textiles whose handling and exposure to contaminant sources during their history is unknown, as discussed in Section 6, and(c)The results of statistical analyses performed on both official and raw data described in Section 3.1 and Section 4.1. These reveal the reasons for the lack of homogeneity of the Shroud data and identify the systematic spatial gradient of the ages as the source of the heterogeneity in means that was not detected by [8], it appears that the conclusion in [8] “*The results provide conclusive evidence that the linen of the Shroud of Turin is medieval*” needs to be reconsidered.

Clearly, the statistical analyses do not unveil the correct age of the Shroud. Our review does not aim to discuss the so-called “authenticity” of the Shroud; we show there are reasons to believe that the 1988 radiocarbon dating does not match current accuracy requirements. Should this be the case, it would be interesting to plan a strategy in order to know the accurate age of the Shroud of Turin.

As pointed out in Section 6, the Shroud was exposed to preservatives during its history, which are potentially able to skew a new ^14^C measurement. In Section 7 we proposed a set of samples to be radiocarbon dated in order to quantify the bias introduced by contaminants not completely removed, or by other sources of heterogeneity such as, for example, mending added to repair the Shroud mentioned in Section 6.1. In detail, we recommended dating the charred yarns of the Shroud, the Holland cloth, the Raes sample, and yarns outside the image from the middle of the Shroud. Taken alone, none of the dating results of these samples could be expected to be the individual source of a reliable estimate of the age of the Shroud. However, the comparison of their ages, in a multivariate analysis, could give selective information on the skewing influence of many of the possible contaminants, as well as their spatial distribution across the cloth.

In summary, radiocarbon measurements along the lines proposed in Section 7 might provide selective information on the impact of several potential contaminants on measurement uncertainty, thus on the most effective precleaning procedure. These data are of basic importance in planning a measurement able to provide accurate dating results, and then to decide if it does make sense to date the Shroud again.

Both the Holland cloth and charred threads were separated from the Shroud after the restoration work in 2002, and are presently conserved in the archbishopric of Turin, together with the Raes sample. Thus, their use would not impact the Shroud.

## Figures and Tables

**Figure 1 entropy-22-00926-f001:**
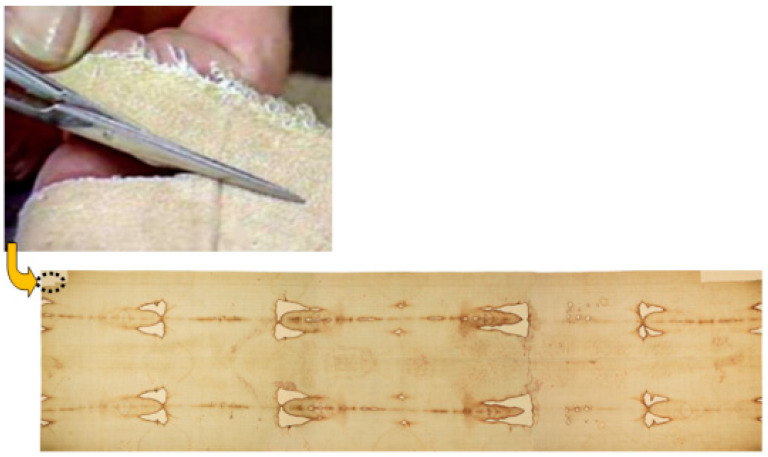
Sample cutting at the bottom left-hand edge of the frontal image of the Shroud.

**Figure 2 entropy-22-00926-f002:**
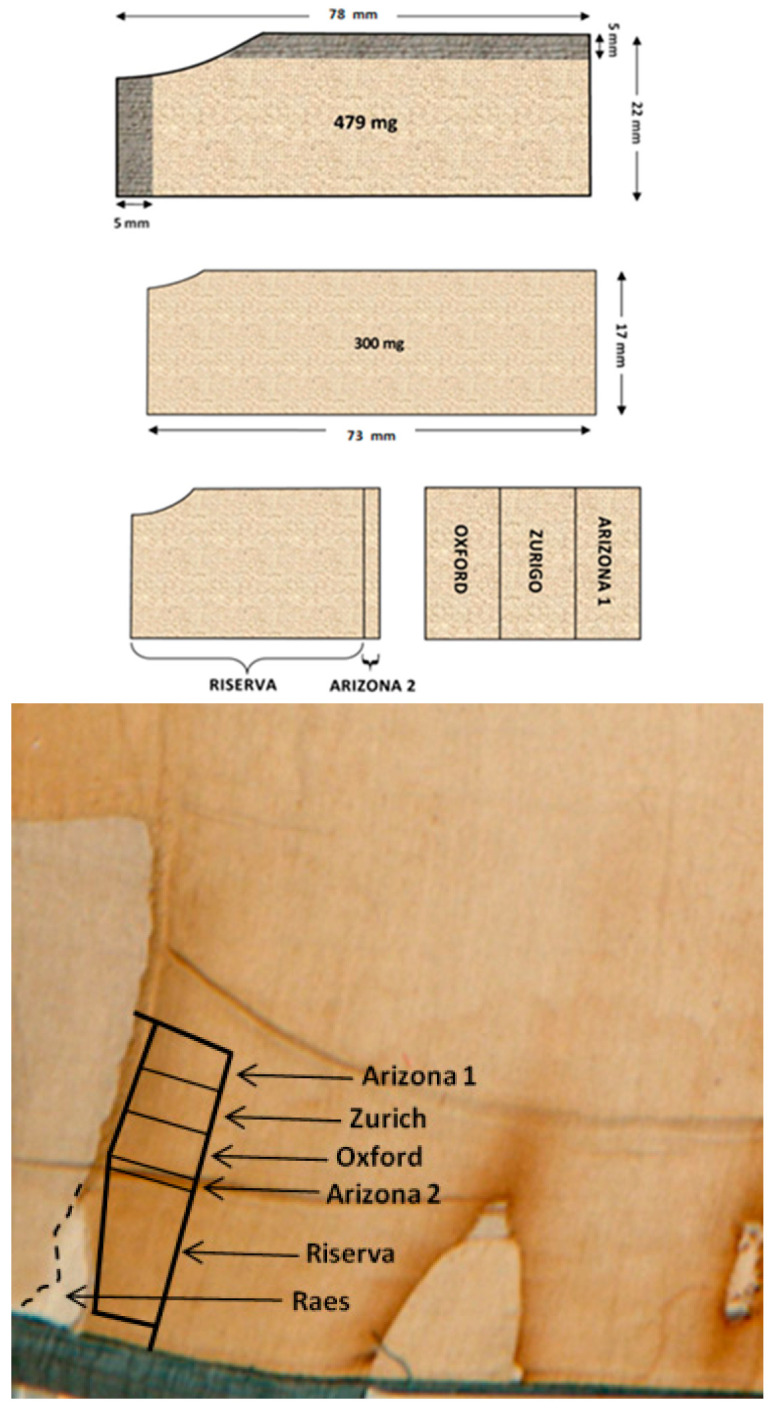
(**Top**): schematic of the Shroud sample to be dated and its initial partition. In the first drawing, the shadowed parts are those trimmed. The second drawing is the part used for the partition. The third drawing shows the retained part, called “Riserva”, on the left and the part to be dated on the right [17]. (**Bottom**): photo of the bottom left-hand edge of the frontal image of the Shroud framed in 1978. (Credit: 1978 Barrie M. Schwortz Collection, STERA Inc., Florissant, CO, USA). We added the subdivisions of the sample and their relative position. The sample removed by Raes in 1973 and the part retained as Riserva are also shown.

**Table 1 entropy-22-00926-t001:** Estimated radiocarbon BP years of the individual subsamples with scaled standard errors from *t*_5_ distribution. Those for Arizona exclude one source of error (see text).

Arizona	RC Dating	591	690	606	701	
	Scaled standard error	30	35	41	33	
Oxford	RC dating	795	730	745		
	Scaled standard error	65	45	55		
Zurich	RC dating	733	722	635	639	679
	Scaled standard error	61	56	57	45	51

**Table 2 entropy-22-00926-t002:** Significance levels of tests of homogeneity of variances and means of the three laboratories for unweighted and weighted analyses.

	Unweighted	Original Weights	Modified Weights
Variance Homogeneity	0.787	0.354	0.700
Difference in Means	0.0400	0.0043	0.0497

**Table 3 entropy-22-00926-t003:** Results of the 1988 dating of eight sub-samples of the Arizona laboratory. On each day, the age of the two sub-samples was normalized with the same standard and blank targets. It is interesting to compare these raw data with those published in [8], see Table 1.

Day	Subsample	Years BP
6 May	A1D(2)	606 ± 41
	A1D(2)’	574 ± 45
12 May	A1D(1)	753 ± 51
	A1D(1)’	632 ± 49
24 May	A1C(1)	676 ± 59
	A1C(1)’	540 ± 57
2 June	A1C(2)	701 ± 47
	A1C(2)’	701 ± 47

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
