# Peer review of "Statistical and Proactive Analysis of an Inter-Laboratory Comparison: The Radiocarbon Dating of the Shroud of Turin"

_entropy, 2020, doi:10.3390/e22090926_

Round 1

Reviewer 1 Report

The problem of radiocarbon dating of the Shroud of Turin is likely to continue until new samples are dated, but even then, there is no full guarantee that everyone will accept these results. Authors try to looking after weak points of the radiocarbon dating method, showing the pre-treatment procedure as a possible reason of the discrepancy. The AAA method is the most common method of chemical procedure used to remove possible contaminants. Its effectiveness depends on the type of material, and is used basically to remove carbonates and humic acid from samples. In case of textiles, wood etc. the common method should be cellulose extraction. Arizona and Oxford added the Soxhlet step into the pre-treatment procedure, which also has potential to remove contaminants soluble in ethanol or pertoleum. As one can see, the Oxford Laboratory has done bleaching which should remove all contamination and leave clean cellulose, and the results seems to be older but still less then 3σ away from the youngest date from Arizona. This shows that the contamination with modern material, if present, is very small.  The isochron on the original material are confusing (what is the age scale AD?) and rather have nothing to do with the reality.  The authors cite the results of the first inter-comparison test [14], but it would be nice to show also results of other tests TIRI, FIRI, VIRI etc. This will show the development of the pre-treatment methods and the quality of the measurements. In line 138 After the dendrochronological calibration ….This is very unprecise and confusing. Dendrochronological dating is used ONLY for trees which forming annual rings. By comparing the sequences of the rings width with the master chronology it is possible to obtain the calendar age of the rings. How you can use it for this textile? I think the authors meant calibrated using radiocarbon calibration curve in this period of time based on radiocarbon concentration in tree rings previously dated dendrochronologically.  After the minor corrections the article could be published. However, I would suggest close work with any radiocarbon laboratory, also those included in the test, to have a overview of the whole radiocarbon dating method.

Reviewer 2 Report

The work is a detiled study of the statistical aspect of the analysis of the Turim Shroud. The work is of interest for a large public, the study is solid and based on consolidated statistical theory.
The manuscropt is well-written and gives a clear exposition of the statistical considerations. The conclusions are complete and consistent with the analysis.
In principle I would say the manuscript can be accept as is stands right now, but I consider that Entropy is not the best journal to have is published, Stats may be a more appropriate MDPI journal.

Reviewer 3 Report

In this paper, di Lazzaro et al. discuss the radiocarbon measurements on the Shroud of Turin, and also make a proposal for new measurements. I believe the paper either needs very major revision or should be rejected in its current form.

A much shorter paper eliminating sections 2-4 to a short summary and arguing the reasons for a re-dating and that proposal seem reasonable to me.

My comments are as follows:

  1. The paper apparently represents the opinions of the authors about the previous dating discussed in reference [7] by Damon et al. (1989). The analysis by [7] is not inaccurate, although one can certainly re-analyze the data and get slight differences. This question has been previously addressed by many authors, including refs. 18, 19, 24, 25, 26, 27, 28, 32 and maybe a few others. I think that the “review” of the statistics of the radiocarbon dates has been beaten to death, indeed even in two previous papers by some of the authors, e.g. refs. 27 and 28. I see no reason to do this again here. The discussion presented in sections 2 and 3 reiterates much of this earlier discussion, in my view adds little to the understanding of the subject and third, does not change the conclusions of ref. [7].
  2. The introduction on the Shroud’s description is biased toward the views presented on the website shroud.com, which is a favour of the “authenticity” of the Shroud as the possible burial cloth of Christ. It ignores detailed scholarship against this “authenticity”, such as presented in the detailed historical work of Nicolotti (2019), The Shroud of Turin: The history and legends of the world’s most famous relic”, Baylor University Press, Waco, TX (English version) and other studies. The alternative scenario that this is a medieval work of art is not really discussed.
  3. In lines 47-52, it is asserted that “this is not the work of an artist”. Only two references supporting this position are given. I would note the historical report (detailed by Nicolotti and others) is that the work appeared around 1353-1354 in a new collegiate church in Lirey (in the Savoy Alps) and its provenance was the subject of detailed discussions at the time, as noted by two bishops of the region. Therefore, the 14C age reported by [7] is completely consistent with the historical record, so there is no reason not to assume it was “the work of an artist”.
  4. Lines 88-89. The statement that the “British Museum was the only referee” is misleading and implies something untoward may have occurred. This needs to be removed and in any case, the term “referee” is inaccurate.
  5. Line 97. Reference [12] is not relevant – it discusses comparing thermoluminescence of pottery and radiocarbon dating.
  6. Figure 2. I think there is a better reference for this diagram. At least, the original drawings of Dr. Riggi should be cited and not the secondary reference from the internet.
  7. Line 129-130. I am not clear as to how the analyses performed later in the paper (using standard methods such as ANOVA) differ from what was available in 1989. ANOVA has been around for a long time.
  8. Lines 142-143. As already noted, this topic has been discussed extensively before. The entire section 3 can be reduced to a brief summary.
  9. Lines 154-155. I have read most of the papers to which the authors refer and I don’t think there is any substantial shift in the age range using the other statistical approaches. The original reference [7] notes the chi-squared for the Shroud sample indicated that the errors might have been underestimated.
  10. Lines 169-173. I don’t think that this conclusion is correct, since the position of the subsamples is not known and was not recorded by the original laboratories.
  11. Lines 236-238. This discussion is model-dependent and assumes one can organize the samples as shown in figure 3. This is just speculation.
  12. Lines 270-275. Reference [32] doesn’t come up with any new interpretations different from the many previous papers (see above) on the statistical analysis. Its main difference is that Casabianca obtained raw data from some of the labs and did a new analysis. No different conclusion was reached, as far as I can tell.
  13. Lines 292-300. Again, what’s new here?
  14. Lines 326-329. You assume here that there are spatial distributions in the cloth. Why would one assume this. If it’s one cloth, it was made at one time on a loom.
  15. Section 6. I agree that today, one would have done a cellulose extraction on all the samples (see Nemec et al. 2010, Radiocarbon 52, 1358-1370. The different pretreatments used in 1988 are not helpful, but I don’t see how they would greatly affect the age, as it is stated in sec. 6.1 that the samples were “relatively clean” [29,43].
  16. Lines 423-424. I think that the samples in [44.45, 47, 48] may be the “Raes” fibres which seem to have a convoluted history and may not, in any case, be clean.
  17. Lines 460-463. You can’t conclude this from the data in ref. [7].
  18. Lines 468-477. I agree here that the recent treatments of the Shroud by the custodian’s team may be problematic.
  19. Section 7.3 I think the Raes fibres at this point have unknown provenance as they have circulated among various groups.
  20. Lines 515-520. A re-analysis and re-measurement is unlikely to show that the Shroud is any different in age than already shown from radiocarbon dating and the historical record.

Round 2

Reviewer 2 Report

The changes do not modify my initial opinion in favor of publication of the manuscript but in another MDPI journal, like Stats.

Reviewer 3 Report

I thank the authors for the revisions and also the extensive response to my comments. I have only a couple of minor comments. I am happy to recommend publication.

  1. Lines 301-304. This statement implies that the British Museum perhaps deliberately withheld information, when in fact, I believe that all the entities concerned had raw data, which was occasionally shared on request with other parties who asked. Perhaps this can be reworded. 
  2. Line 305. Although "serious incongruities" is a quote of ref. [32], I am not sure it is needed here. 
  3. I thank the authors for section 3.2 which clarifies the "spatial trend" discussion. 
  4. As far as the proposal in section 7 and 8 is concerned, I wish the authors luck in obtaining the samples. However, if the authors do obtain the samples, I expect that the results will not end the discussion of the provenance of the Shroud, as is amply demonstrated in its history, see ref. [9].